# RETAINING BY DOING: THE ROLE OF ON-POLICY DATA IN MITIGATING FORGETTING

## ABSTRACT

Adapting language models (LMs) to new tasks via post-training carries the risk of degrading existing capabilities—a phenomenon classically known as *catastrophic forgetting*. In this paper, we set out to identify guidelines to mitigate this phenomenon, by systematically comparing the forgetting patterns of two widely adopted post-training methods: supervised fine-tuning (SFT) and reinforcement learning (RL). Our experiments reveal a consistent trend across LM families (Llama, Qwen) and tasks (instruction following, general knowledge, and arithmetic reasoning): RL leads to less forgetting than SFT while achieving comparable or higher target task performance. To investigate the cause for this difference, we consider a simplified setting in which the LM is modeled as a mixture of two distributions, one corresponding to prior knowledge and the other to the target task. We identify that the *mode-seeking* nature of RL, which stems from its use of *on-policy* data, enables keeping prior knowledge intact when learning the target task. We then verify this insight by demonstrating that the use on-policy data underlies the robustness of RL to forgetting in practical settings, as opposed to other algorithmic choices such as the KL regularization or advantage estimation. Lastly, as a practical implication, our results highlight the potential of mitigating forgetting using *approximately* on-policy data, which can be substantially more efficient to obtain than fully on-policy data.

## 1 INTRODUCTION

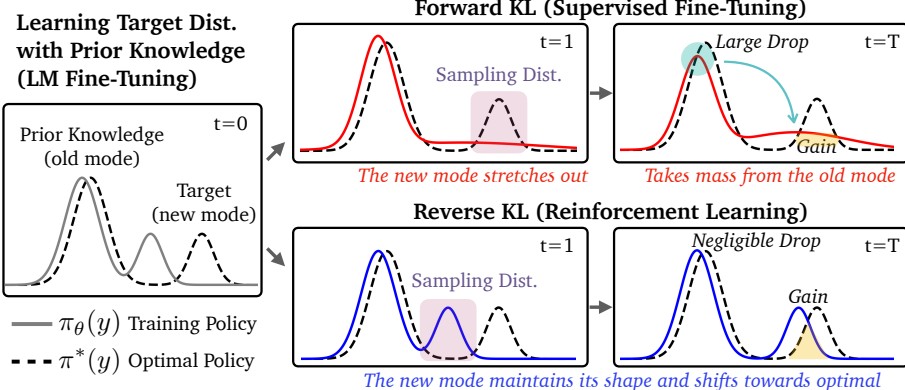

Figure 1: Illustration of the forgetting dynamics for the forward KL objective, corresponding to SFT, and the reverse KL objective, corresponding to RL. Left: we model LM post-training as a mixture of two modes. The "old" mode represents prior knowledge and the "new" mode represents a target task. Initially, the old mode of the training policy $\pi_\theta$ roughly matches the old mode of the optimal policy $\pi^*$, but its additional "new" mode does not match the new target mode. The goal is for the training policy to match the optimal policy. Top right: minimizing forward KL first stretches the new mode of $\pi_\theta$ and then moves probability mass from the old mode to cover the target, leading to forgetting. Bottom right: in contrast, minimizing reverse KL maintains the shape of the old mode and covers the target distribution by shifting the new mode of $\pi_\theta$.

Adapting language models (LMs) to new target tasks during post-training carries the risk of eroding previously acquired capabilities—a phenomenon known as *catastrophic forgetting* (McCloskey & Cohen, 1989; Kirkpatrick et al., 2017). Such forgetting has been reported to occur when training LMs to follow instructions via supervised fine-tuning (SFT) (Luo et al., 2023; Shi et al., 2024; Wu et al., 2024) or aligning them with human preferences via reinforcement learning (RL) (Bai et al., 2022; Ouyang et al., 2022). However, the understanding of how SFT and RL compare in terms of their susceptibility to forgetting remains limited.

In this work, we systematically compare the forgetting patterns of SFT and RL in order to identify principled guidelines for mitigating forgetting in LM post-training. We conduct a comprehensive study across instruction following, general knowledge, and arithmetic reasoning tasks, using Qwen 2.5 (Yang et al., 2024) and Llama 3 (Grattafiori et al., 2024) models of up to 8B scale. Our experiments reveal a consistent trend: SFT suffers from severe forgetting, whereas RL can achieve high target task performance without substantial forgetting (Figure 2).

We then investigate the cause for the relative robustness of RL to forgetting. At first glance, it may seem at odds with conventional wisdom. Namely, minimizing the cross-entropy loss via SFT is equivalent to minimizing the *forward KL* divergence with respect to the optimal policy, while maximizing the RL objective corresponds to minimizing the *reverse KL* (Korbak et al., 2022a). Conventional wisdom presumes that the *mode-seeking* nature of reverse KL enables faster learning of target distributions (Chan et al., 2022; Vaswani et al., 2022; Tajwar et al., 2024b) at the cost of losing coverage of old modes, while the *mode-covering* forward KL should maintain probability mass across modes. We reconcile this discrepancy by considering a simplified setting, where the target distribution is modeled as a mixture of two distributions: one representing the policy's prior knowledge and the other representing the target task. We show that, if the initial policy is uni-modal (*i.e.*, has a single mode), then SFT can in fact be more robust than RL to forgetting. However, if the initial policy is multi-modal (*i.e.*, has multiple modes), which is arguably the case for practical LMs, then mode-seeking RL leads to less forgetting than mode-covering SFT; see Figure 1 for an illustration.

The mode-seeking behavior of RL (*i.e.*, its accordance with reverse KL minimization) stems from the usage of *on-policy* data. Through extensive ablations, we empirically verify that this property underlies the robustness of RL to forgetting, as opposed to other algorithmic choices such as the advantage estimation or the application of KL regularization. Moreover, we explore what degree of on-policy data allows mitigating forgetting. We find that for SFT, while generating data only from the initial policy is not enough, *approximately on-policy* data generated at the start of each epoch can suffice for substantially reducing forgetting. This suggests a practical guideline for LM post-training: leveraging on-policy data, potentially sampled asynchronously or at the start of each epoch for improved efficiency, can reduce unintended disruption of the model's existing capabilities.

To summarize, our main contributions are:

- We demonstrate that RL is more robust to forgetting than SFT through extensive experiments on instruction following, general knowledge, and reasoning tasks, using LMs from different families and scales.

- We provide intuition for why the mode-seeking nature of RL, which stems from its use of on-policy data, can counterintuitively lead to less forgetting than mode-covering SFT.

- We corroborate this insight by demonstrating that the use of on-policy data underlies the robustness of RL to forgetting in practical settings, and highlight the potential of mitigating forgetting through approximately on-policy data, which can be substantially more efficient to obtain than fully on-policy data.

## 2 FORGETTING IN LM POST-TRAINING

We begin by introducing notation and the metrics used to measure forgetting. Then, we empirically compare the forgetting patterns of supervised fine-tuning (SFT) and reinforcement learning (RL).

## 2.1 PRELIMINARIES

A language model (LM) is modeled by a policy $\pi_\theta(y \mid x)$, where the response $y$ is generated conditioned on the prompt $x$. For a target task $\mathcal{T}$, we denote the optimal policy by $\pi^*(\cdot \mid x)$. In SFT, the cross-entropy loss is minimized with respect to ground truth responses $y^*$ sampled from the optimal policy: $\mathcal{L}_{\text{SFT}}(\theta; x) := \sum_y -\pi^*(y \mid x) \log \pi_\theta(y \mid x)$. By contrast, in RL, the goal is to maximize the KL-regularized reward with respect to responses generated by the LM and a reward function $r(x, y) \in \{0, 1\}$[1] *i.e.*: $J_{\text{RL}}(\theta; x) := \mathbb{E}_{y \sim \pi_\theta(\cdot \mid x)}[r(x, y)] - \beta \cdot \text{KL}[\pi_\theta(\cdot \mid x) \, \| \, \pi_{\theta_0}(\cdot \mid x)]$, where $\beta > 0$ and $\pi_{\theta_0}$ is the initial policy.

**Forgetting and evaluation metrics.** The initial policy $\pi_{\theta_0}$ is trained on a target task $\mathcal{T}$ for $T$ optimization steps, resulting in the trained policy $\pi_{\theta_T}$. This policy is evaluated using accuracy, which measures the fraction of correct outputs generated by $\pi_{\theta_T}$ for prompts associated with $\mathcal{T}$. We denote the accuracy of $\pi_{\theta_T}$ over $\mathcal{T}$ by $\mathcal{A}(\pi_{\theta_T}, \mathcal{T})$ and define the *target task gain* as $\Delta_g := \mathcal{A}(\pi_{\theta_T}, \mathcal{T}) - \mathcal{A}(\pi_{\theta_0}, \mathcal{T})$. We quantify forgetting, based on on a collection of tasks $\{\mathcal{T}'_j\}_{j=1}^M$, through the *non-target tasks drop* $\Delta_d := \frac{1}{M} \sum_{j=1}^M \mathcal{A}(\pi_{\theta_0}, \mathcal{T}'_j) - \mathcal{A}(\pi_{\theta_T}, \mathcal{T}'_j)$. During post-training, the aim is to achieve high target task gain while minimizing as much as possible the non-target tasks drop. For brevity, we will often refer to target task gain as *gain* and to non-target tasks drop as *drop*.

## 2.2 EXPERIMENTAL SETUP

**Target tasks and evaluation.** We consider three tasks, covering different capabilities: IFEval (Zhou et al., 2023) for instruction following, MMLU (Hendrycks et al., 2021) for general knowledge, and Countdown (Pan et al., 2025) for arithmetic reasoning. After training on one target task, we evaluate the model's performance on all the other tasks. We additionally include as non-target tasks two safety datasets, WildJailbreak (Jiang et al., 2024) and WildGuardTest (Han et al., 2024), since safety capabilities are often eroded through fine-tuning (Qi et al., 2024), making them highly suitable for measuring forgetting. In our RL experiments, correct generations are assigned a reward of 1 and incorrect generations are assigned a reward of 0.

**Models and baselines.** We use instruct models from the Llama 3 (Grattafiori et al., 2024) and Qwen 2.5 (Yang et al., 2024) families: Llama-3.2-1B-Instruct, Llama-3.1-8B-Instruct, Qwen-2.5-1.5B-Instruct, and Qwen-2.5-7B-Instruct. We compare two SFT variants and one RL method: **1) SFT**, which uses responses generated by Llama-3.3-70B-Instruct as ground truth responses; **2) Self-SFT**, which uses responses generated by the initial model, and **3) GRPO**, which is a common RL algorithm (Shao et al., 2024) for tasks with verifiable outputs. For both SFT variants, the generated data was filtered using the reward function to include only examples with correct responses. We use Self-SFT as a baseline to represent the typical first step in the post-training pipeline when human labels are absent (Dong et al., 2023; Lambert et al., 2024). All models are trained for two epochs. Additional implementation details are provided in Appendix A.3.

## 2.3 RESULTS: SFT FORGETS MORE THAN RL

Figure 2 compares the target task gain and non-target tasks drop of the SFT variants and GRPO. We observe higher levels of forgetting in SFT compared to GRPO across datasets, model families, and sizes. In particular, we find:

- For Self-SFT, achieving a similar target accuracy gain to GRPO induces a significantly larger drop on non-target tasks.

- While SFT can achieve a higher performance gain than GRPO on the instruction following task, it induces an even larger drop on non-target tasks relative to Self-SFT.

- As shown in Figure 3, a high learning rate is typically required to reach high target performance for SFT, often at the cost of severe forgetting; a smaller learning rate reduces forgetting but fails to reach the same target performance even with more epochs.

Overall, both SFT variants exhibit a consistent tradeoff between target performance and forgetting, whereas GRPO improves target performance without noticeable drops on non-target tasks.

---

[1] We use RL with verifiable reward (RLVR) throughout our experiments.

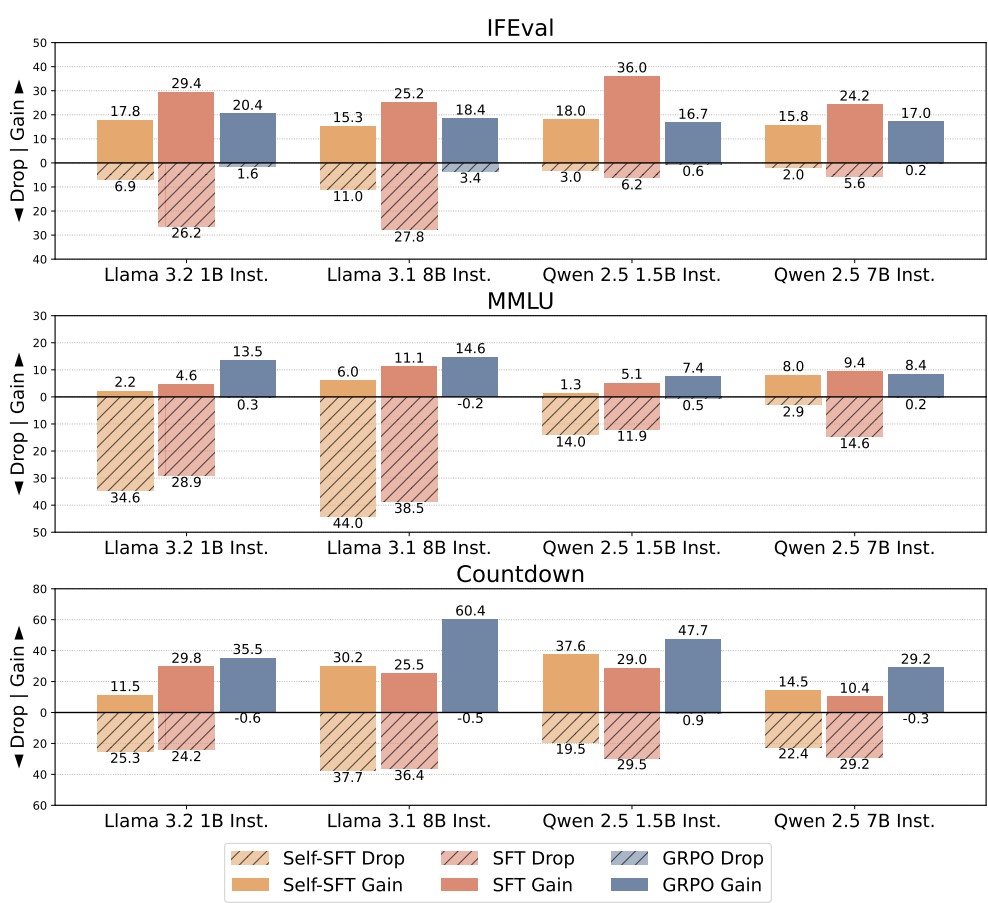

Figure 2: **SFT forgets more than RL across tasks and models.** We compare the **Gain** (solid bar) / **Drop** (shaded bar) across models and datasets for: *(1)* Self-SFT, which uses data generated from the initial policy; *(2)* SFT, which uses data generated by Llama-3.3-70B-Instruct; and *(3)* GRPO. Gain (higher better) represents the accuracy increase on the target task, while drop (lower better) represents the average accuracy decrease on non-target tasks.

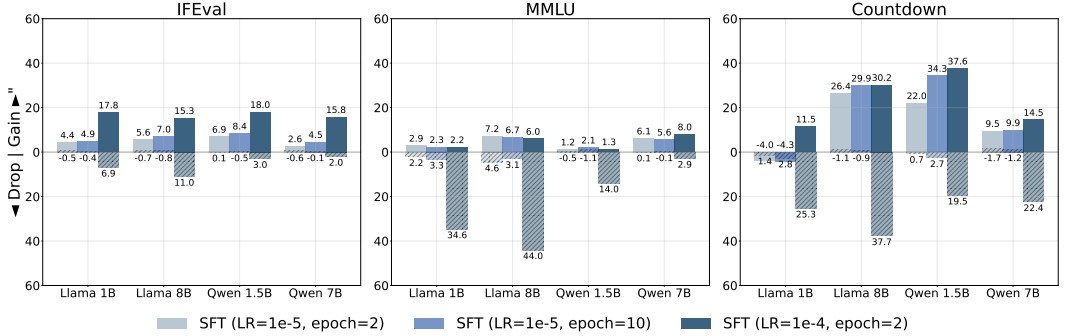

Figure 3: **SFT exhibits a tradeoff between target task performance and forgetting.** Comparison of SFT runs with different learning rates ($1e-5, 1e-4$) and training epochs $(2, 10)$.

## 3 UNDERSTANDING FORGETTING DYNAMICS THROUGH THE LENS OF KL

SFT and RL can be viewed as minimizing different directions of the KL divergence with respect to the optimal policy. Specifically, as reviewed in §3.1, SFT corresponds to forward KL minimization

while RL corresponds to reverse KL minimization. Intuitively, a *mode-seeking* objective such as reverse KL should be more susceptible to forgetting: it moves probability mass quickly from one mode to another, whereas *mode-covering* forward KL should better maintain probability mass on all modes. This intuition is invalidated in light of the evidence presented in §2.3, showing that SFT causes more forgetting than RL. We address this discrepancy through an empirical analysis of a simplified setting with univariate Gaussian distributions. The analysis reveals that SFT can in fact lead to less forgetting than RL if the initial policy has a single mode (§3.2). However, in more realistic multi-modal scenarios, RL is more robust to forgetting (§3.3).

## 3.1 SFT AND RL AS KL MINIMIZATION

**SFT as forward KL minimization (mode-covering).** It is widely known that SFT is equivalent to minimizing the forward KL between the optimal and training policies since:

$$\mathcal{L}_{\text{SFT}}(\theta; x) = \sum_y -\pi^*(y \,|\, x) \log \pi_\theta(y \,|\, x) = \text{KL}\big[\pi^*(\cdot \,|\, x) \,\|\, \pi_\theta(\cdot \,|\, x)\big] + \mathcal{H}(\pi^*(\cdot \,|\, x)),$$

where $\mathcal{H}(\pi^*(\cdot \,|\, x))$ is the entropy of $\pi^*(\cdot \,|\, x)$, which does not depend on $\pi_\theta$.

**RL as reverse KL minimization (mode-seeking).** The optimal policy for the KL-regularized RL objective (§2.1) is given by $\pi^*(y \,|\, x) = \frac{1}{Z(x)} \pi_0(y \,|\, x) \exp(r(x, y)/\beta)$ (Korbak et al., 2022a), where $\pi_0$ is the initial policy, $Z(x) := \sum_y \pi_0(y \,|\, x) \exp(r(x, y)/\beta)$ is the partition function, and $\beta > 0$ is the KL regularization coefficient. This implies that one can view the maximization of the RL objective as minimization of the reverse KL from $\pi^*$ since:

$$J_{\text{RL}}(\theta; x) = \mathbb{E}_{y \sim \pi_\theta(\cdot \,|\, x)}[r(x, y)] - \beta \cdot \text{KL}[\pi_\theta(\cdot \,|\, x) \,\|\, \pi_0(\cdot \,|\, x)]$$
$$= -\beta \cdot \text{KL}[\pi_\theta(\cdot \,|\, x) \,\|\, \pi^*(\cdot \,|\, x)] + \beta \cdot \log Z(x),$$

where $\ln Z(x)$ does not depend on $\pi_\theta$ (*c.f.* Korbak et al. (2022a); Tajwar et al. (2024a)).

## 3.2 FORWARD KL FORGETS LESS IN A UNI-MODAL SETTING

In this section, we demonstrate that forward KL (SFT) leads to less forgetting than reverse KL (RL) under a uni-modal training policy. We model the optimal policy as a mixture of two univariate Gaussian distributions to mirror LM fine-tuning: an "old" mode that corresponds to prior knowledge and a "new" mode that represents the target task. As shown below, results in this setting align with the intuition stated at the beginning of the section, by which the mode-covering forward KL should forget less. However, in the next section we show that once the uni-modal training policy is expanded to a multi-modal one, reverse KL causes less forgetting.

**Setup.** The optimal policy is modeled by an "old" mode representing prior knowledge and a "new" mode representing a target task:

$$\pi^*(y) = \alpha^* \cdot p_{\text{old}}(y; \theta^*_{\text{old}}) + (1 - \alpha^*) \cdot p_{\text{new}}(y; \theta^*_{\text{new}}), \tag{1}$$

where $\alpha^* \in (0, 1)$ and the distributions $p_{\text{old}}$ and $p_{\text{new}}$ are univariate Gaussians with means and standard deviations given by $\theta^*_{\text{old}} = (\mu^*_{\text{old}}, \sigma^*_{\text{old}})$ and $\theta^*_{\text{new}} = (\mu^*_{\text{new}}, \sigma^*_{\text{new}})$, respectively. In this section, the training policy $\pi_\theta$ is modeled as a univariate Gaussian with trainable mean $\mu$ and standard deviation $\sigma$, *i.e.*, $\theta = (\mu, \sigma)$. We define the target task gain and non-target task drop as the change in *overlap area*[2] between the training policy and the modes of the optimal policy. Concretely, the overlap area for the old and new modes is defined as:

$$S_{\text{old}}(\theta) := \frac{\int_{-\infty}^{\infty} \min\{\alpha^* p_{\text{old}}(y), \pi_\theta(y)\} \, dy}{\alpha^*} \,, \; S_{\text{new}}(\theta) := \frac{\int_{-\infty}^{\infty} \min\{(1 - \alpha^*) p_{\text{new}}(y), \pi_\theta(y)\} \, dy}{1 - \alpha^*}. \tag{2}$$

Notice that $S_{\text{old}}(\theta), S_{\text{new}}(\theta) \in [0, 1]$. The target task gain at training step $T$ is accordingly defined by $\Delta_g := S_{\text{new}}(\theta_T) - S_{\text{new}}(\theta_0)$ and the non-target tasks drop is $\Delta_d := S_{\text{old}}(\theta_0) - S_{\text{old}}(\theta_T)$. We initialize the training policy $\pi_\theta$ such that it covers the mode of $\pi^*$ corresponding to $p_{\text{old}}$, and compare minimizing the forward and reverse KL objectives (defined in §3.1) with respect to $p_{\text{new}}$ in terms of their gain-drop tradeoff. The parameters in $\theta$ are updated through sample-based gradients,

---

[2]The overlap area can be formulated via the total variation distance; see Appendix A.2.

where for forward KL data is sampled from $p_{new}$ and for reverse KL it is sampled from $\pi_\theta$. See Appendix A.1 for additional implementation details.

**Results.** Figure 4 shows the forgetting patterns of forward and reverse KL. To reach a target task gain of $0.9$, forward KL results in a non-target tasks drop of $0.64$ while reverse KL leads to a larger drop of $0.7$. This matches common intuition: the mode-covering forward KL stretches the probability mass to cover the new mode while retaining more mass on the old mode compared to the mode-seeking reverse KL. That is, in this setting, forward KL causes less forgetting than reverse KL.

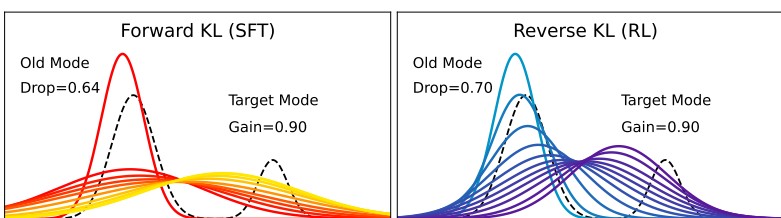

Figure 4: **Forward KL (SFT) with uni-modal training policy forgets less than reverse KL (RL).** Learning and forgetting dynamics of forward KL (left) and reverse KL (right). Dashed lines represent the modes of the optimal policy: $p_{old}$ (left) and $p_{new}$ (right). For forward KL, the curve goes from red to yellow as training progresses. For reverse KL, the curve goes from blue to purple. Forgetting corresponds to the decrease of overlap on the left mode and learning a new target task corresponds to the increase in overlap on the right mode.

### 3.3 REVERSE KL FORGETS LESS IN A MULTI-MODAL SETTING

We showed in §3.2 that the mode-covering properties of forward KL (SFT) lead to less forgetting than reverse KL (RL) when the initial training policy is uni-modal. This stands in contrast to the experiments of §2.3, which show that in practical LM post-training settings, RL is more resilient to forgetting. In this section, we reconcile this discrepancy by showing that when we allow the initial training policy to have multiple modes, arguably a closer match to practice, the mode-seeking reverse KL results in less forgetting.

**Setup.** We consider the setup of §3.2, where the optimal policy is modeled as a mixture of two Gaussian distributions (Equation 1). Instead of modeling the training policy as a uni-modal Gaussian, we now model it as a bi-modal distribution:

$$\pi_\theta(y) = \alpha \cdot q_{old}(y; \theta_{old}) + (1 - \alpha) \cdot q_{new}(y; \theta_{new}), \qquad (3)$$

where $\theta = (\alpha, \theta_{old}, \theta_{new})$ is the trainable parameters, with $\alpha \in [0, 1]$ being a mixture weighting, $\theta_{old} = (\mu_{old}, \sigma_{old})$ defining the mean and standard deviation of a univariate Gaussian $q_{old}$, and $\theta_{new} = (\mu_{new}, \sigma_{new})$ similarly defining a univariate Gaussian $q_{new}$. We initialize the training policy $\pi_\theta$ such that $q_{old}$ roughly covers the mode of $\pi^*$ corresponding to $p_{old}$ and, as in §3.2, compare the gain-drop tradeoffs exhibited by forward and reverse KL minimization with respect to $p_{new}$. See Appendix A.1 for additional implementation details.

**Results.** Figure 5 shows that, with a high learning rate, achieving a target task gain of $0.9$ with forward KL causes severe forgetting—the area overlap with $p_{old}$ drops by $0.12$. Reducing the learning rate can mitigate forgetting of the old mode, but leads to failure in learning the target $p_{new}$. By contrast, reverse KL shifts $q_{new}$ toward $p_{new}$ while largely keeping the old mode intact. This simulation demonstrates that, for bi-modal policies, reverse KL can match a new target mode without redistributing probability mass from a mode that represents prior knowledge.

## 4 LEARNING FROM ON-POLICY DATA MITIGATES FORGETTING

The experiments of §2 demonstrated that RL causes less forgetting than SFT. By considering a simplified setting in §3, we identified that the mode-seeking behavior of RL, which stems from its usage of on-policy data, may underlie its robustness to forgetting. In this section, we verify this prospect and explore the following natural question: *what degree of on-policy data allows mitigating forgetting?* As evident from the results of Self-SFT in Figure 2, generating data only from the

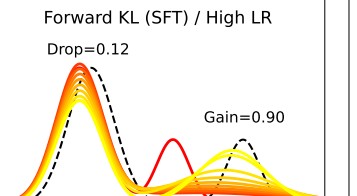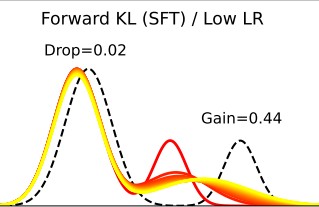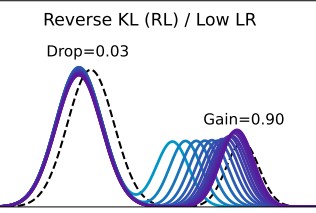

Figure 5: **Reverse KL (RL) with multi-modal training policy forgets less than forward KL (SFT).** Learning and forgetting patterns of forward KL with different high (0.15) and low (0.01) learning rates (left and middle) and reverse KL (right). Dashed lines represent the modes of the optimal policy: $p_{\text{old}}$ (left) and $p_{\text{new}}$ (right). For forward KL, the curve goes from red to yellow as training progresses. For reverse KL, the curve goes from blue to purple. Forgetting corresponds to the decrease of overlap on the left mode and learning a new task corresponds to the increase in overlap on the right mode.

initial policy is not enough. However, in §4.1 we show that SFT with approximately on-policy data, generated at every epoch or with on-policy traces produced by RL, can suffice for substantially reducing forgetting. These findings, along with additional experiments in §4.2, strongly indicate that the robustness of RL to forgetting arises from its use of on-policy data, as opposed to other algorithmic choices such as the use of KL regularization or an advantage estimation (in contrast to a hypothesis made in concurrent work (Lai et al., 2025)).

## 4.1 APPROXIMATELY ON-POLICY DATA CAN SUFFICE FOR MITIGATING FORGETTING

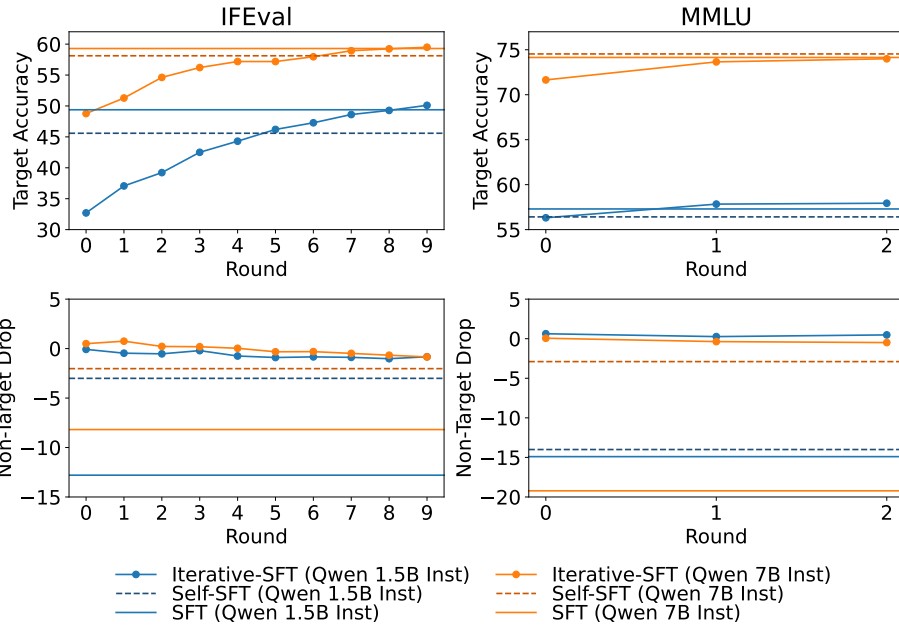

Figure 6: **Approximately on-policy data can suffice for mitigating forgetting in SFT.** This figure compares target task accuracy (top) and average non-target accuracy drop (bottom) for three SFT variants: *(1)* Iterative-SFT, which uses data generated at the start of each round (*i.e.*, epoch); *(2)* Self-SFT, which uses data generated from the initial policy; and *(3)* SFT, which uses fully off-policy data generated by a separate expert model (Llama-3.3-70B-Instruct). While SFT and Self-SFT suffer from severe forgetting, our results show that using approximately on-policy data, generated at the start of each epoch, can suffice for mitigating forgetting (Iterative-SFT).

**SFT using approximately on-policy data.** Figure 2 showed that Self-SFT, which generates data only from the initial policy, suffers from severe forgetting. On the other hand, RL, which generates

data at every step and thus represents the most on-policy end of the spectrum, is robust to forgetting. We now test whether Iterative-SFT, an *approximately* on-policy approach that iteratively trains on data generated at the start of each epoch, can suffice for mitigating forgetting (Zelikman et al., 2022; Dong et al., 2023; Xiong et al., 2025). Specifically, in Figure 6 we compare the target task accuracy and the drop in non-target tasks of Iterative-SFT, Self-SFT, and SFT. We find that Iterative-SFT is able to reach a target accuracy that is higher than or comparable to that of SFT, while only exhibiting mild to no forgetting. We also test an additional approximately on-policy approach that applies SFT on data generated during an RL run, and observe reduced forgetting as well (see Appendix A.4.2). Overall, these results highlight that while RL remains most effective in forgetting mitigation, making SFT more on-policy or directly applying SFT on RL data can suffice for reducing forgetting.

**The advantage estimator in GRPO does not underlie its robustness to forgetting.** Concurrent work (Lai et al., 2025) attributed the robustness of GRPO to forgetting to an implicit regularization of the advantage estimator. The fact that SFT on approximately on-policy data does not suffer from forgetting (as shown above) stands in contrast to this hypothesis. In Appendix A.4.1, we provide further support for on-policy data, and not any particular choice of advantage estimate, being the main factor mitigating forgetting by demonstrating that RL without an advantage estimator (*i.e.,* REINFORCE (Williams, 1992)) is also robust to forgetting.

## 4.2 KL REGULARIZATION DOES NOT EXPLAIN ROBUSTNESS TO FORGETTING

KL regularization is commonly applied during RL to prevent the policy from drifting too far from its initialization (Ouyang et al., 2022; Shao et al., 2024). We examine whether this regularization accounts for the lesser forgetting of RL. As Figure 7 shows, non-regularized GRPO achieves a similar target task gain and non-target tasks drop tradeoff as KL-regularized GRPO across all considered models and datasets, except for Llama models trained on IFEval. These results suggest that the use of KL regularization does not underlie the robustness of RL to forgetting.

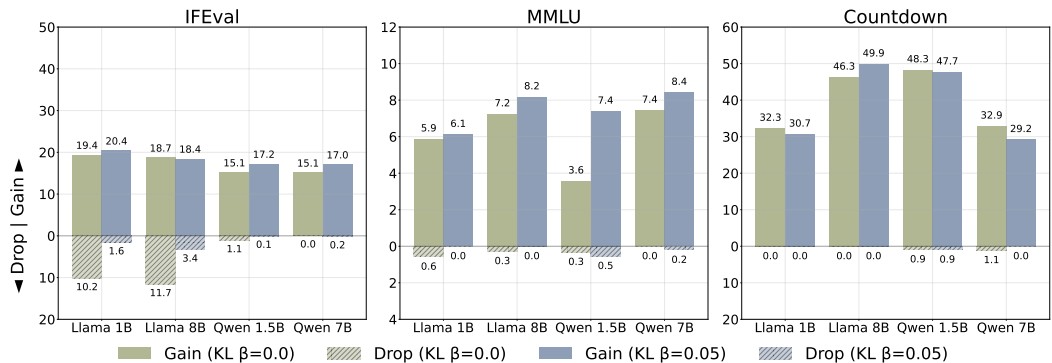

Figure 7: **KL regularization is not a major contributor to RL's lesser degree of forgetting.** Comparison of GRPO with KL regularization ($\beta = 0.05$) and without it ($\beta = 0.0$). Except for the Llama model family and IFEval target task, the non-regularized GRPO achieves a similar gain-drop tradeoff as regularized GRPO.

## 5 RELATED WORK

**Catastrophic forgetting.** Catastrophic forgetting has been studied since the early research on connectionist models (McCloskey & Cohen, 1989). Many early studies focus on preventing parameters to drastically change (Kirkpatrick et al., 2017; Li & Hoiem, 2018; Lopez-Paz & Ranzato, 2017). In modern language models post-training, patterns of catastrophic forgetting differ due to the massive data learned during pre-training (Luo et al., 2023; Shi et al., 2024; Wu et al., 2024). While LMs typically do not drastically forget all the pre-trained knowledge, post-training LMs still lead to degradation in performance, which has been called "alignment tax" (Bai et al., 2022; Ouyang et al., 2022). Studies found more severe forgetting in domains that easily interfere such as instruction following and safety (Qi et al., 2023; He et al., 2024). Recent studies discover that the observed forgetting in LMs may be spurious—the forgotten behaviors or abilities can be revived with little

training on the data from the forgotten distribution (Lee et al., 2024; Chen et al., 2024). Kotha et al. (2024) posited that forgetting happens when the LM infers a wrong mode from the mixture of distributions to perform the task, and a carefully selected prompt can recover forgetting. Our work draws inspiration from the mixture of distributions perspective to establish intuition.

**LM post-training.** Post-training techniques such as SFT and RL are widely adapted to endow the pretrained language models with desired behaviors or enhancements in target domains. SFT relies on the ground truth responses provided in the datasets. RL allows the model to generate its own responses and only provide the reward at the end be it from a parameterized reward model (Schulman et al., 2017) or a verifiable reward module (Shao et al., 2024; Lambert et al., 2024). Recent studies have shown that SFT and RL exhibit distinct characteristics. Chu et al. (2025) showed that RL is able to generalize to unseen distributions while SFT mostly memorizes. (Wang et al., 2025) observed that RL can benefit from even training on a single example without severe overfitting. (Mukherjee et al., 2025) reported that RL naturally fine-tunes a smaller network compared to SFT. A common thread connecting these observations is that the parameter update during RL training is more local and targeted. Other methods such as RAFT (Dong et al., 2023) and STaR (Zelikman et al., 2022) performs SFT in several rounds and can be viewed as on-policy offline RL. This paper complements these studies and provide a forgetting-centric view on the difference between SFT and RL.

**Forward KL vs reverse KL.** Studies have connected SFT and RL through the lens of KL divergence. Tajwar et al. (2024a) showed that optimizing the policy gradient objective in RL can be viewed as minimizing the reverse KL between the training policy and the optimal policy (Korbak et al., 2022b), which exhibits a mode-seeking behavior to fit the target distribution efficiently. In contrast, SFT typically uses the cross-entropy loss, which implements the forward KL objective. This perspective has been used to preserve diversity in LM's generation (Li et al., 2025). While the intuition that forward KL discourages mode collapse can be found from prior RL work such as Vaswani et al. (2022); Chan et al. (2022), which analyze how forward KL objectives encourage maintaining support over multiple modes rather than collapsing onto a single high-reward mode, the notion of forgetting have not been made explicit.

**Concurrent work.** Similarly to our work, Lai et al. (2025); Shenfeld et al. (2025) have concurrently found that RL exhibits less forgetting than SFT. However, Lai et al. (2025) attribute RL's robustness to an implicit regularization of a particular advantage estimator. We provide evidence against this claim in §4, and instead identify the crucial role of on-policy data in mitigating forgetting. Shenfeld et al. (2025) also highlight the benefits of on-policy data through a perspective that is complementary to ours (§3). Though, our work goes beyond fully on-policy data and demonstrates the potential of approximately on-policy data in more efficiently mitigating forgetting.

## 6  CONCLUSION

We systematically compared catastrophic forgetting in SFT and RL for LM post-training. Across tasks, scales, and model families, we found that RL consistently achieves strong target performance with substantially less forgetting than SFT. Our experiments in both simplified and real-world settings establish that the robustness of RL to forgetting primarily stems from its use of on-policy data, rather than other algorithmic choices such as the advantage estimate or KL regularization. Furthermore, they highlight the potential of efficiently mitigating forgetting by incorporating approximately on-policy data, sampled asynchronously or at the start of each epoch.

**Limitations and future directions.** Our work provides evidence that RL is more robust than SFT to forgetting across several tasks, model families, and scales. However, investigating how forgetting patterns vary as the model and dataset sizes are further scaled, beyond our compute budget, remains a valuable direction for future work. Moreover, while we provide intuition for why RL forgets less than SFT based on a simplified mixture-of-Gaussians setting (§3) and empirically identify the use of on-policy data as a main cause for this difference in forgetting (§4), additional research is necessary to theoretically establish the role of on-policy data in mitigating forgetting. Going forward, the issue of forgetting becomes central as the community moves toward building agents that continually learn from experience (Silver & Sutton, 2025). Deciding what data to consume is consequential to the stability of the agent. Our results indicate that data generated on-policy will better preserve existing capabilities, and is therefore safer to learn from, than off-policy data such as content on the internet or experience from other agents. In a similar vein, our insights lays groundwork for understanding forgetting in the emerging paradigm of test-time training (Sun et al., 2020; Hardt & Sun, 2024).

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

## A    ADDITIONAL EXPERIMENTS AND IMPLEMENTATION DETAILS

### A.1    SIMULATION DETAILS

**Setup.**    We use a univariate mixture-of-Gaussians synthetic task to compare forward KL (FKL; SFT analogue) update against reverse KL (RKL; RL analogue). We calculate gradient updates using $n = 1000$ samples. For evaluation and plots, densities are computed on a uniform grid at every 100 iterations.

**Single-mode setting.**    We run the gradient step updates for $T = 1000$ iterations or when the target task gain reaches 0.9. The training policy starts as a single-mode univariate Gaussian at the old mode, initialized as $\mathcal{N}(-3.2, 1.0)$ (75% old mass), and is adapted toward the same target mixture used above: $0.75 \cdot \mathcal{N}(-3.0, 1.0) + 0.25 \cdot \mathcal{N}(3.5, 0.7)$. We use an FKL learning rate 0.05 and a RKL learning rate 0.05.

**Bi-modal setting.**    We run the gradient step updates for $T = 1000$ iterations or when the target task gain reaches 0.9. The initial policy $\pi_\theta(x)$ is a two-component mixture with weight 0.75 on an "old" Gaussian $\mathcal{N}(-3.5, 1.0)$ and 0.25 on a "new" Gaussian $\mathcal{N}(0.5, 0.7)$. The target $\pi^*(x)$ is a mixture with the same weights over $\mathcal{N}(-3.0, 1.0)$ (old) and $\mathcal{N}(3.5, 0.7)$ (new). We sweep two FKL learning rates $\{0.15, 0.01\}$ and use a RKL learning rate 0.01.

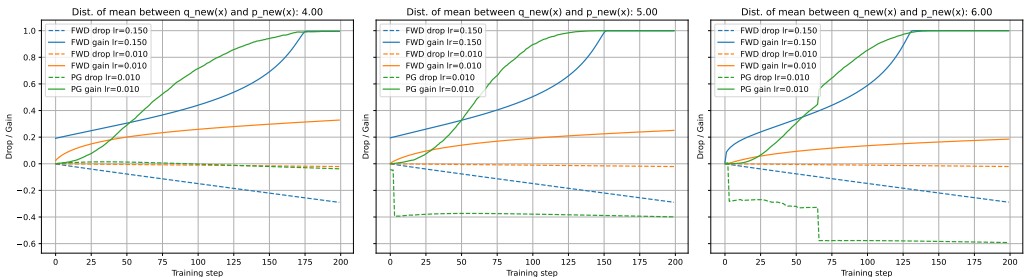

Figure 8: Simulation comparison with different distances ($[4.0, 5.0, 6.0]$) between $p_{\text{new}}$ and $q_{\text{new}}$.

**RL forgets when the target distribution is far.**    We show in Figure 8 the simulation results with varying distance ($4.0 - 6.0$) between $q_{\text{new}}$ and $p_{\text{new}}$. We observe that as the distance gets larger, RL begins to suffer from forgetting as well. suggesting that RL is not immune to forgetting when the target task is drastically far away from the starting modes.

### A.2    CONNECTION BETWEEN AREA OVERLAP AND TOTAL VARIATION DISTANCE

Let $f, g : \mathcal{D} \to \mathbb{R}_{\geq 0}$ be, possibly unnormalized, integrable density functions over a domain $\mathcal{D}$. The total variation (TV) between $f$ and $g$ is defined by:

$$\text{TV}(f, g) \; := \; \frac{1}{2} \int_{\mathcal{D}} |f(y) - g(y)| dy \,.$$

Notice that:

$$\int_{\mathcal{D}} \min\{f(y), g(y)\} \, dy = \int_{\mathcal{D}} \frac{1}{2} \big(f(y) + g(y) - |f(y) - g(y)|\big) dy$$

$$= \frac{1}{2} \left( \int_{\mathcal{D}} f(y) dy + \int_{\mathcal{D}} g(y) dy \right) - \frac{1}{2} \int_{\mathcal{D}} |f(y) - g(y)| dy \qquad (4)$$

$$= \frac{1}{2} \left( \int_{\mathcal{D}} f(y) dy + \int_{\mathcal{D}} g(y) dy \right) - \text{TV}(f, g) \,.$$

Now, in the context of §3.2, recall that the *area overlap* of the training policy $\pi_\theta$ with respect to the old mode of the optimal policy is defined by (Equation (2)):

$$S_{\text{old}}(\theta) = \frac{\int_{-\infty}^{\infty} \min\{\alpha^* p_{\text{old}}(y), \pi_\theta(y)\} \, dy}{\alpha^*} \,.$$

Choosing $f = \alpha^* p_{\text{old}}$ and $g = \pi_\theta$, by Equation (4) we may write $S_{\text{old}}(\theta)$ as follows:

$$S_{\text{old}}(\theta) = \frac{\frac{1}{2}(\alpha^* + 1) - \text{TV}(\alpha^* p_{\text{old}}, \pi_\theta)}{\alpha^*} = \frac{1}{2} + \frac{1}{2\alpha^*} - \frac{1}{\alpha^*} \text{TV}(\alpha^* p_{\text{old}}, \pi_\theta).$$

Hence, the non-target tasks drop at training step $T$ is equal to the normalized increase in total variation distance between the training policy and the (scaled) old component of the optimal policy:

$$\Delta_d = S_{\text{old}}(\theta_0) - S_{\text{old}}(\theta_T) = \frac{\text{TV}(\alpha^* p_{\text{old}}, \pi_{\theta_T}) - \text{TV}(\alpha^* p_{\text{old}}, \pi_{\theta_0})}{\alpha^*}.$$

Similarly, the area overlap of the training policy $\pi_\theta$ with respect to the new mode of the optimal policy is given by:

$$S_{\text{new}}(\theta) = \frac{\frac{1}{2}(\alpha^* + 1) - \text{TV}(\alpha^* p_{\text{old}}, \pi_\theta)}{\alpha^*} = \frac{1}{2} + \frac{1}{2(1 - \alpha^*)} - \frac{1}{1 - \alpha^*} \text{TV}((1 - \alpha^*) p_{\text{new}}, \pi_\theta).$$

This implies that the target task gain at training step $T$ is equal to the normalized decrease in total variation distance between the training policy and the (scaled) new component of the optimal policy:

$$\Delta_g = S_{\text{new}}(\theta_T) - S_{\text{new}}(\theta_0) = \frac{\text{TV}((1 - \alpha^*) p_{\text{new}}, \pi_{\theta_T}) - \text{TV}((1 - \alpha^*) p_{\text{new}}, \pi_{\theta_0})}{1 - \alpha^*}.$$

### A.3 TRAINING DETAILS

**Training details.** We used the AdamW optimizer. The learning rate was initialized to $1e{-}4$ for Llama-3.2-1B-Instruct and Qwen-2.5-1.5B-Instruct and $5e{-}6$ for Llama-3.1-8B-Instruct and Qwen-2.5-7B-Instruct. We use cosine scheduler with warp-up step ratio $0.03$ over the course of training. Each model was trained with a batch size of $128$ for IFEval and MMLU and $64$ for Countdown. Unless otherwise specified, training was run for 2 epochs.

For supervised fine-tuning (SFT), we minimized the cross-entropy loss with a maximum sequence length of $4096$. For reinforcement learning (RL) experiments, we used the GRPO algorithm with a KL-penalty coefficient of $0.05$ and apply updates right after the group samples are generated (hence no advantage clipping). All experiments were implemented in PyTorch and trained on maximally 8 H100 GPUs with mixed-precision (bfloat16) training.

### A.4 EXTRA EXPERIMENTS AND ABLATIONS

#### A.4.1 REINFORCE IS AS ROBUST AS GRPO TO FORGETTING

| Model | Method | IFEval | | MMLU | | Countdown | |
|-------|--------|--------|--------|--------|--------|--------|--------|
| | | Gain | Drop | Gain | Drop | Gain | Drop |
| Llama 3.1 8B Inst. | REINFORCE | 17.8 | 7.7 | 8.6 | -0.1 | 7.5 | -0.8 |
| | GRPO | 18.4 | 3.4 | 14.6 | -0.2 | 60.4 | -0.5 |
| Qwen 2.5 7B Inst. | REINFORCE | 5.7 | 2.9 | 6.4 | -0.6 | 11.9 | -0.1 |
| | GRPO | 17.0 | 0.2 | 8.4 | 0.2 | 29.2 | -0.3 |

Table 1: Gain and drop comparison between GRPO and REINFORCE.

In this appendix, we compare GRPO with REINFORCE (Williams, 1992), a classical policy gradient RL algorithm that does not employ an advantage estimator. Table 1 shows that REINFORCE lags behind GRPO in optimizing the target task accuracy, yet maintains a similar low level of forgetting. This suggests that algorithmic differences, such as the advantage estimator used in RL, primarily affect the magnitude of performance gains, whereas the mitigation of forgetting can be primarily attributed to the use of on-policy data.

### A.4.2 SFT USING RL TRACES.

Data generated by RL throughout training is on-policy with regard to the model at each optimization step. When this RL data is later used for SFT, the process moves away from being fully on-policy, though it remains distinct from fully off-policy approaches such as SFT. We investigate whether SFT on RL data can also mitigate forgetting. In Figure 9, we observe that SFT trained on RL (GRPO) data trails full RL marginally in terms of gains but exhibits only slightly larger forgetting. This highlights a yet-to-be-identified benefit of using RL data for SFT (DeepSeek-AI et al., 2025).

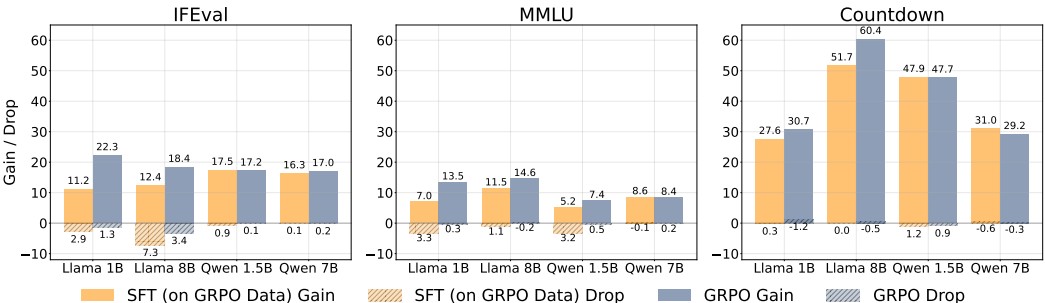

Figure 9: **SFT over on-policy traces produced by RL exhibits reduced forgetting.** This plot shows the comparison between SFT trained on GRPO data and GRPO.

### A.4.3 OFF-POLICY DATA

|  |  | IFEval | | MMLU | | Countdown | |
|---|---|---|---|---|---|---|---|
|  |  | Gain ↑ | Drop ↓ | Gain ↑ | Drop ↓ | Gain ↑ | Drop ↓ |
| Llama 3.1 8B Instruct | SFT w/ GRPO on-policy data | 12.4 | 7.3 | 11.5 | 1.2 | 51.7 | 0.0 |
|  | SFT w/ Qwen data | 17.9 | 9.0 | 10.4 | 2.5 | 47.9 | 0.9 |
| Qwen 2.5 7B Instruct | SFT w/ GRPO on-policy data | 16.3 | 0.1 | 8.6 | -1.4 | 31.0 | -0.6 |
|  | SFT w/ Llama data | -3.0 | 9.8 | 7.6 | 7.2 | 10.1 | 6.6 |

Table 2: SFT results on data generated from a different model.

Table 2 further shows that the on-policy data only works for the model that generates it. Training on another model's on-policy data increases forgetting.

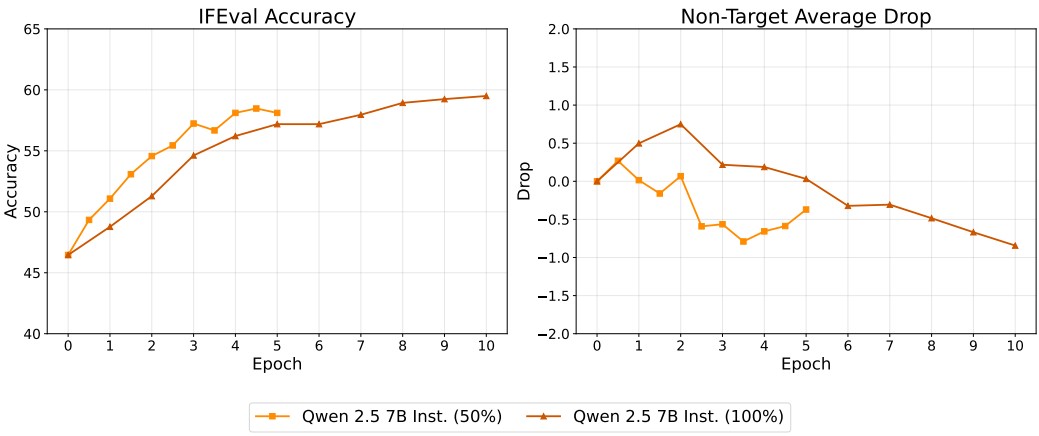

Figure 10: **Fraction of approximately on-policy data still mitigates forgetting.** The figure compares Iterative-SFT on IFEval when varying the fraction of dataset prompts for which the model generates outputs before SFT (50% vs. 100%). Left: accuracy for IFEval as the target task. Right: non-target tasks drop.

### A.4.4 FRACTION OF THE APPROXIMATELY ON-POLICY DATA

In Figure 10, we vary the amount of on-policy data (50%, and 100%) generated before running SFT. On IFEval, we observe that Qwen-2.5-7B-Instruct trained with Iterative-SFT with higher update frequency (50%) improves task performance more rapidly. While the drop is slightly more rapid for 50%, it plateaus quickly and stays within a similar range as 100%.

## A.5 KL DIVERGENCE FROM INITIAL POLICY AND FORGETTING CORRELATE MODERATELY

|  |  | IFEval | | MMLU | | Countdown | |
| --- | --- | --- | --- | --- | --- | --- | --- |
|  |  | Drop (%) | KL | Drop (%) | KL | Drop (%) | KL |
| Llama-3.2-1B-Instruct | Self-SFT | 6.9 | 52.4 | 34.6 | 39.3 | 25.3 | 796.7 |
|  | SFT | 26.2 | 61.1 | 28.9 | 48.4 | 24.2 | 1254.7 |
|  | GRPO | 1.6 | 2.6 | 0.3 | 4.1 | -0.6 | 70.6 |
| Qwen-2.5-1.5B-Instruct | Self-SFT | 3.0 | 25.0 | 14.0 | 3.0 | 19.5 | 896.1 |
|  | SFT | 6.2 | 47.8 | 11.9 | 9.2 | 29.5 | 846.6 |
|  | GRPO | 0.6 | 1.5 | 0.5 | 0.4 | 0.9 | 34.9 |

Table 3: Non-target tasks drop and $\mathrm{KL}[\pi_{\theta_0} \| \pi_\theta]$ for the Llama-3.2-1B-Instruct and Qwen-2.5-1.5B-Instruct models from Figure 2.

Shenfeld et al. (2025) have concurrently found that RL exhibits less forgetting than SFT, attributing RL's robustness to an implicit regularization of RL toward policies with low KL divergence from the initial policy $\pi_{\theta_0}$. In particular, Shenfeld et al. (2025) empirically identify $\mathrm{KL}[\pi_{\theta_0} \| \pi_\theta]$ as an indicator for the extent of forgetting, mostly through an extensive evaluation on synthetic tasks. We explore the connection between $\mathrm{KL}[\pi_{\theta_0} \| \pi_\theta]$ and forgetting in the setup of §2, where the KL divergence is estimated based on 100 examples from the evaluation set. Table 3 shows that, in accordance with the hypothesis of Shenfeld et al. (2025), GRPO exhibits both smaller KL divergences and smaller drops in non-target tasks performance compared to the SFT variants. Furthermore, the Pearson correlation between KL divergence and non-target tasks drop across all models, methods, and datasets is 0.52. Yet, when comparing Self-SFT and SFT, the relation between KL divergence and forgetting is less monotonic—a larger KL does not necessarily imply a higher degree of forgetting. This indicates that the relationship between KL divergence and forgetting is still not fully understood.

## A.6 LLM USE IN THE PAPER

We use LLM (ChatGPT) to aid writing. In particular, after drafting a sentence we use the LLM to "Make it read better." or "Make it read more fluently." We also use LLM to help format the references and find the right source and venues that they are published in. We validate the links and details provided by the model to the best of our ability.

