# OpenReview forum: "Retaining by Doing: The Role of On-Policy Data in Mitigating Forgetting"
_ICLR.cc/2026/Conference — Submitted to ICLR 2026_

### Official Review · Reviewer_uGfs · 2025-10-23

**Soundness:** 3
**Presentation:** 3
**Contribution:** 3
**Rating:** 6
**Confidence:** 4

**Summary:**

This paper systematically compares catastrophic forgetting in language models during post-training via supervised fine-tuning (SFT) and reinforcement learning (RL). Through experiments across multiple model families and tasks, the authors demonstrate that RL exhibits significantly less forgetting than SFT while achieving comparable or better target task performance. They attribute this robustness to RL's mode-seeking behavior, which stems from its use of on-policy data, and further show that even approximately on-policy data can mitigate forgetting in SFT.

**Strengths:**

1. The paper provides a comprehensive and rigorous empirical evaluation across diverse tasks, making the findings highly robust and generalizable.
2. The authors offer an intuitive yet formal explanation for the observed phenomenon by modeling the policy as a mixture of distributions and linking forgetting behavior to the mode-seeking nature of reverse KL minimization.
3. The practical implication that approximately on-policy data can significantly reduce forgetting is a valuable and efficient alternative to full on-policy RL, offering a useful guideline for real-world model adaptation.

**Weaknesses:**

1. Forgetting is measured via average accuracy drops; other forms of degradation (semantic drift, safety loss, calibration changes) are not quantitatively explored.
2. The experiments are limited to models of up to 8B parameters, and it is unclear whether the same trends hold for significantly larger or smaller models, limiting the scalability claims.
3. While the Gaussian mixture analogy provides valuable intuition, it may oversimplify the complex, high-dimensional, and often non-Gaussian nature of real-world language model distributions.

**Questions:**

1. How closely must data match the current policy to be considered "approximately on-policy," and what are the precise thresholds for its effectiveness in mitigating forgetting?
2. Does the observed robustness of RL hold for capabilities beyond knowledge and reasoning, such as in multimodal or conversational tasks, where forgetting patterns might differ?
3.  Is the reduced forgetting in RL achieved at the cost of slower convergence or higher sample complexity compared to SFT, and what are the implied trade-offs for practical deployment?

---

> ### Author Response · Authors · 2025-11-24
> **Response to uGfs**
>
> We thank the reviewer for the thoughtful feedback. We appreciate the reviewer’s recognition of the soundness and comprehensiveness of our experiments, and the the importance and practical implications of our investigation. We address the reviewer’s questions below.
>
> **Appropriateness of evaluation and the metrics**
>
> The reviewer questioned the comprehensiveness of evaluation and whether accuracy is sufficient.
>
> > Forgetting is measured via average accuracy drops
>
> While we agree that forgetting is multi-faceted, we argue that our evaluation follows established practice in the continual learning literature. Based on the continual learning survey [1] section 3.2, our target task gains capture overall performance similar to Average Accuracy (AA) and our non-target task drops resemble Forgetting Measure (FM). In either case, the canonical evaluations are based on task accuracy.
>
> [1] A Comprehensive Survey of Continual Learning: Theory, Method and Application by Wang et al.
>
>
> > Other forms of degradation (semantic drift, safety loss, calibration changes) are not quantitatively explored.
>
> First, we would like to point out that our evaluation includes forgetting on safety using WildGuardTest and WildJailbreak as described in Section 2.2. In addition, we provide extra evaluation on the KL between the base model and the fine-tuned model to capture distributional drift in Appendix A.5.
>
> If the reviewer has specific alternative metrics, we are happy to consider it!
>
> **Model scale and generality of our claims**
>
> > The experiments are limited to models of up to 8B parameters, and it is unclear whether the same trends hold for significantly larger or smaller models, limiting the scalability claims.
>
> While we agree that scaling the model size could further strengthen our findings, we are constrained by the amount of compute available to afford experiments beyond 8B. We choose 8B as the standard model to fine-tune in academic settings. We do include model sizes of 1B in both the Llama and Qwen model families, and the trend that SFT forgets more than RL holds for all these variations (Section 2.2, Figure 2). We believe that the current range of model families and sizes are reasonably sufficient to support our empirical findings.
>
>
>
> **How closely must data match the current policy to be considered "approximately on-policy," and what are the precise thresholds for its effectiveness in mitigating forgetting?**
>
> In Figure 10 and Appendix A.4.4 in our revision, we vary the fraction of on-policy data (50%, and 100% of the entire dataset) generated before running SFT.
> On IFEval, we observe that Qwen-2.5-7B-Instruct trained with Iterative-SFT with higher update frequency (50%) improves task performance more rapidly.
> While the drop is slightly more for 50%, it plateaus quickly and stays within a similar range as 100%, showing a similar degree of forgetting when changing the "on-policyness" by 2x. Due to the compute constraint, we aren't able to perform a full sweep. We will include a wider range sweep in the updated version.
>
>
> (Response 1/2)

---

> ### Author Response · Authors · 2025-11-24
> **Response to uGfs**
>
> **Evaluation beyond reasoning and knowledge**
>
> > Does the observed robustness of RL hold for capabilities beyond knowledge and reasoning, such as in multimodal or conversational tasks, where forgetting patterns might differ?
>
> We include extra experiments on AlpacaEval (conversational task) as suggested, and show that the benefit of RL in mitigating forgetting is consistent with the main experiments in our paper. While evaluating on multimodal models could provide additional validation, our investigation is anchored around text based language models, and we believe our evaluation across knowledge, reasoning, safety, and the additional conversational tasks provide solid evidence to support our claim that RL forgets less than SFT.
>
> Below we show the different runs evaluated on AlcapaEval, a benchmark for chat-based evaluation. We report WR (win rate in %) against GPT-4 following standard practice.
>
> |            | Initial Model | SFT on IFEval | RL on IFEval |
> | --------------------- | ----------- | ------------- | ------------ |
> | Llama-3.1-8B-Instruct | 41.6        | 23.5          | 43.1         |
> | Qwen2.5-7B-Instruct   | 36.5        | 22.5          | 35.2         |
>
> |                   | Initial Model | SFT on MMLU  | RL on MMLU |
> | --------------------- | ----------- | ----------- | ---------- |
> | Llama-3.1-8B-Instruct | 41.6        | 11.6        | 41.2       |
> | Qwen2.5-7B-Instruct   | 36.5        | 19.9        | 36.0       |
>
> |                       | Initial Model | SFT on Countdown | RL on Countdown |
> | --------------------- | ----------- | ---------------- | --------------- |
> | Llama-3.1-8B-Instruct | 41.6        | 0.0              | 42.0            |
> | Qwen2.5-7B-Instruct   | 36.5        | 0.4              | 34.6            |
>
> We observe that for both Llama-3.1-8B-Instruct and Qwen-2.5-7B-Instruct, SFT suffers much more after training on the three different target tasks. On the other hand, we observe almost no drop or even improved performance for Llama-3.1-8B-Instruct after RL training compared to the base policy WR, and only very mild drop for Qwen-2.5-7B-Instruct after RL. The results are consistent with the other evaluations reported in the paper.
>
>
>
>
> **Is the reduced forgetting in RL achieved at the cost of slower convergence or higher sample complexity compared to SFT, and what are the implied trade-offs for practical deployment?**
>
> In our setup, reduced forgetting in RL does not come from slower convergence or seeing more distinct training prompts. SFT, Self-SFT, and RL are all trained for the same number of epochs on the same target-task prompts, with matched optimization hyperparameters. Under this matched setting, RL reaches comparable or higher target-task performance than SFT while forgetting less on non-target tasks, so the effect is not due to RL simply being “under-trained” or converging more slowly.
> The main trade-off for practical deployment is compute per update. RL requires generating responses before applying policy updates, which increases the compute requirement. This is precisely why we study approximately on-policy SFT variants such as Iterative-SFT: by refreshing data from the current policy only periodically, they substantially reduce forgetting relative to fully off-policy SFT, while incurring much smaller overhead than full RL. Practically, this yields a spectrum: off-policy SFT is cheapest but forgets the most; full RL is most robust but most expensive; and approximate on-policy SFT sits in the middle, recovering much of RL’s robustness at a more favorable compute–forgetting trade-off.
>
> **The Gaussian setting is limited**
>
> The reviewer pointed out that the Gaussian mixture analysis is a stylized setting and may not fully capture the complexity of high-dimensional LMs.
> We agree that the mixture-of-Gaussians analysis is not a full theory of forgetting in LLMs and we do not intend to present it as such, and its role in the paper is more modest. It isolates one key ingredient: optimizing reverse KL with respect to a target mode while a separate “old” mode is already well modeled, and shows that under this structure, mode-seeking behavior _can_ actually mitigate forgetting instead of exacerbating it, like in the unimodal case.
> This mirrors the LM setting where a pretrained model already captures a rich “prior knowledge” distribution and post-training adds a new mode corresponding to a target task.
>
>
> (Response 2/2)

---

### Official Review · Reviewer_rYjr · 2025-10-24

**Soundness:** 3
**Presentation:** 4
**Contribution:** 2
**Rating:** 4
**Confidence:** 4

**Summary:**

The paper investigates catastrophic forgetting of tow prevalent LLM post-training paradigms: supervised fine-tuning and reinforcement learning. Empirical results across several model families and tasks show that RL (particularly on-policy algorithms like GRPO) achieves comparable or better target-task performance while exhibiting less forgetting on non-target tasks. To explain this, the authors propose that RL’s mode-seeking behavior (reverse KL), rooted in its use of on-policy data, helps preserve prior knowledge. A simplified mixture-of-Gaussians analysis further illustrates how SFT and reverse RL behave differently under uni-modal vs multi-modal assumptions.

**Strengths:**

- The paper articulates a concrete and important question, i.e., why RL fine-tuning forgets less than SFT, and provides extensive experimental results supporting the finding across architectures and datasets.

- The writing and figures are well-organized. In particular, the “gain–drop” metric and visualization (Figure 2) make results intuitive, and the toy Gaussian analysis offers a didactic explanation.

- The study disentangles potential confounders (KL regularization, advantage estimation) and clearly isolates the effect of data policy.

**Weaknesses:**

- The finding that on-policy learning mitigates forgetting better than off-policy learning has already been explored in both RL and alignment literature. Notably, “Preference Fine-Tuning of LLMs Should Leverage Suboptimal, On-Policy Data” (Tajwar et al., 2024) also frames on-policy vs off-policy updates as mode-seeking vs mode-covering, drawing the same connection between reverse KL and improved retention. Thus, while this paper extends that reasoning to an explicit forgetting study, its conceptual contribution over the established framework appears incremental.

- Given that on-policy methods (iterative SFT, DPO, rejection-sampling, GRPO, PPO) are already standard practice, the practical insight, i.e., “on-policy mitigates forgetting”, may have limited influence on future method design unless coupled with deeper theoretical grounding or new algorithmic proposals.

- The mixture-of-Gaussians setting helps intuition but does not rigorously establish the link between KL directionality, multimodality, and empirical forgetting in high-dimensional LLMs.

**Questions:**

I noticed that the statement “Conventional wisdom presumes that the mode-seeking nature of reverse KL enables faster learning … while the mode-covering forward KL should maintain probability mass across modes.” cites (Chan et al., 2022; Tajwar et al., 2024b). However, upon reviewing Tajwar et al. (2024b), I couldn’t find an explicit discussion of the latter claim regarding forward KL preserving mode coverage. Could the authors please clarify whether this interpretation is directly supported by that work or derived from general understanding in the literature?

---

> ### Author Response · Authors · 2025-11-24
> **Response to rYjr**
>
> We thank the reviewer for the thoughtful feedback. We appreciate the reviewer’s recognition of the importance of our investigation, the soundness and comprehensiveness of our experiments, and the clarity of our ideas presented. We address the reviewer’s questions below.
>
> **Contribution and novelty of our finding**
>
> > on-policy learning mitigates forgetting better than off-policy learning has already been explored in both RL and alignment literature. Notably, Tajwar et al., 2024 also frames on-policy vs off-policy updates as mode-seeking vs mode-covering, drawing the same connection between reverse KL and improved retention.
>
> We would like to clarify our contribution with regard to the findings in Tajwar et al., 2024.
> First, Tajwar et al., 2024 does _not_ tackle the issue of forgetting, which has been the main focus of our study. They use forward KL (mode-covering) and reverse KL (mode-seeking) to understand the condition for efficient learning of target distributions. Specifically, they theoretically show that reverse KL moves the probability mass more aggressively. “[...] reverse KL can quickly re-distribute probability mass to only a subset of the required categories likely in target distribution, within a few gradient steps.”
>
> Extrapolating from their observations, one would postulate a trend that contradicts our empirical finding that SFT forgets less than RL, because aggressive redistribution of probability implies a more drastic degradation in the previous high probability region. While Tajwar et al., 2024 did not make this point, and we formalize and demonstrate this explicitly in Section 3.2, showing that reverse KL leads to more forgetting in a unimodal case (matching the intuition of Figure 1 in Tajwar et al., 2024). We then show that this is _not_ the full picture: with the presence of an “old” mode (analogous to pretrained prior), reverse KL actually leads to less forgetting (Section 3.3), matching our empirical finding.
> While we take inspiration from Tajwar et al., 2024 to use mode-seeking/covering as tools for the analysis, our insights and investigations are original.
>
>
>
>
> **Actionable insights from our investigation**
>
> > Given that on-policy methods (iterative SFT, DPO, rejection-sampling, GRPO, PPO) are already standard practice, the practical insight, i.e., “on-policy mitigates forgetting”, may have limited influence on future method design unless coupled with deeper theoretical grounding or new algorithmic proposals.
>
> We would like to briefly reiterate our contributions: 1) showing the trend “SFT forgets more than RL” through comprehensive empirical evidence, 2) establishing the intuition that the reverse KL objective can lead to less forgetting when the “old” mode is present, showing the benefit of on-policy data for mitigating forgetting, and 3) examining the core elements of on-policy data and found that even the less costly “approximately” on-policy can lead to a lesser degree of forgetting.
>
> Although we do not introduce new algorithms, our study enhances our understanding of the consequences of forgetting when applying existing methods. To our knowledge, we are the first to make the connection between on-policyness and forgetting explicit (with concurrent work discussed in the Related Work section), and we believe these insights are useful for guiding future algorithm design. For example, our findings can inform method selection under cost constraints: when full RL is too expensive, one might prefer Iterative-SFT over standard SFT to retain some of the forgetting-mitigation benefits of on-policy training.
>
>
>
>
> **Forward KL preserves mode coverage**
>
> The reviewer pointed out that Tajwar et al. (2024) does not explicitly state that forward KL preserves mode coverage. The intuition that forward KL discourages mode collapse can be found from prior RL work such as Vaswani et al. (2022) [1] and Chan et al. (2022) [2], which analyze how forward KL objectives encourage maintaining support over multiple modes rather than collapsing onto a single high-reward mode.
>
> In our paper, this intuition is made explicit in Section 3.2. In a uni-modal training-policy setting, we show that minimizing forward KL leads to smaller drops in probability mass on the “old” mode than minimizing reverse KL at matched target gains. This is the statement that forward KL better maintains coverage of the existing mode and thus induces less forgetting in that regime. We then extend this analysis to the multi-modal training-policy case in Section 3.3, where the conclusion inverts and reverse KL can better preserve the old mode, which is the regime we argue is closer to LLM post-training.
>
> We have incorporated this clarification into the related section in the revision.
>
> [1] A general class of surrogate functions for stable and efficient reinforcement learning by Vaswani et al., 2022
>
> [2] Greedification Operators for Policy Optimization: Investigating Forward and Reverse KL Divergences by Chen et al., 2022
>
> (1/2)

---

> > ### Author Response · Authors · 2025-12-03
> > **Response to rYjr**
> >
> > **The Gaussian setup is limited**
> >
> > The Reviewer pointed out that the Gaussian mixture analysis is a stylized setting and may not fully capture the complexity of high-dimensional LMs.
> > We agree that the mixture-of-Gaussians analysis is not a full theory of forgetting in LLMs and we do not intend to present it as such, and its role in the paper is more modest. It isolates one key ingredient: optimizing reverse KL with respect to a target mode while a separate “old” mode is already well modeled, and shows that under this structure, mode-seeking behavior _can_ actually mitigate forgetting instead of exacerbating it, like in the unimodal case.
> > This mirrors the LM setting where a pretrained model already captures a rich “prior knowledge” distribution and post-training adds a new mode corresponding to a target task.
> >
> > (2/2)

---

### Official Review · Reviewer_toGA · 2025-10-29

**Soundness:** 4
**Presentation:** 3
**Contribution:** 3
**Rating:** 6
**Confidence:** 4

**Summary:**

This paper studies the problem of catastrophic forgetting in the post-training of LLMs. It finds that reinforcement learning suffers almost no forgetting because it uses on-policy data, whereas SFT easily forgets due to off-policy training. The authors propose Iterative-SFT, which generates new data with the current model at the beginning of each epoch, achieving approximately on-policy learning. This method avoids the complexity of RL training while significantly mitigating forgetting.

**Strengths:**

- The paper conducts both experimental evaluations and theoretical analyses of catastrophic forgetting in SFT and RL, and the conclusions are convincing.
- It clearly shows that the reason RL resists catastrophic forgetting lies in its on-policy nature of data, rather than KL regularization or advantage estimation, which is an observation of notable value.

**Weaknesses:**

- The practicality of Iterative-SFT may be limited for two reasons: 1) Since the policy model generates its own training data, the generated examples may not be sufficiently challenging compared to data produced by a stronger teacher model; 2) It requires a reward model or rule-based verification methods to score and filter the data; however, because the policy model itself may not be well-versed in the target domain, the proportion of high-quality samples could be low, placing high demands on the reward model/rule-based verification methods.

**Questions:**

- Why are the Self-SFT and SFT curves shown as straight lines, rather than plotted after each epoch like Iterative-SFT with a stepwise curve?

---

> ### Author Response · Authors · 2025-11-24
> **Response to toGA**
>
> We thank the reviewer for the thoughtful feedback. We appreciate the reviewer for recognizing our contribution especially in its importance, the comprehensiveness and soundness of our findings. We clarify the raised questions below.
>
> **Why are the Self-SFT and SFT curves shown as straight lines, rather than plotted after each epoch like Iterative-SFT with a stepwise curve?**
>
> In all our main experiments, Self-SFT and SFT are trained for 2 epochs with fixed hyperparameters, so the curves appear as straight lines. By contrast, Iterative-SFT is not run for a predetermined number of epochs. Instead, we repeat the process: generate new samples using the current model on the entire dataset and fine-tune on these samples, until its final target performance matches that of SFT. This adaptive stopping criterion is important because our goal is to compare forgetting at matched target-task gain, not at a fixed number of epochs. Once we fix the gain level, plotting per-epoch checkpoints for Iterative-SFT naturally results in a stepwise curve.
>
>
> **The practicality of Iterative-SFT**
>
> The reviewer pointed out two potential limitations of Iterative-SFT: 1) the need for a verifiable reward function, and 2) the model's original ability might be limited compared to a strong teacher. We provide clarification on the method as well as the role of Iterative-SFT in our paper.
>
> First, we would like to clarify the setting: reliance on the verifiable reward function is the same for Iterative-SFT, SFT, and RL. In SFT, the teacher’s generations are also filtered by the reward function; in RL, the reward function is an integral part of the algorithm. That is, Iterative-SFT does not introduce extra assumptions regarding the use of the reward function compared to the other methods.
>
> On the second point, we agree with the reviewer that Iterative-SFT is limited by the base policy’s ability, which is the same way RL is limited by its base policy. The main difference between Iterative-SFT and RL is how much data the policy generates before performing a model update—higher “on-policyness” (model update frequency) means faster learning on the new task. The flip side is that frequent updates are costly, so we explore whether Iterative-SFT, a less on-policy alternative, can provide the same benefit in mitigating forgetting conditioning on the desired performance being reached. In Figure 6, we ensure that Iterative-SFT matches SFT in the target accuracy and compare the degree of forgetting.
>
> The role of Iterative-SFT in our paper is to study catastrophic forgetting by varying the degree of “on-policyness” from full RL, as opposed to proposing it as an off-the-shelf new post-training alternative.

---

### Author Response · Authors · 2025-12-03
**Rebuttal Summary**

We sincerely thank all reviewers and chairs for their time and effort in providing valuable comments and suggestions. To facilitate assessment, we offer the following rebuttal summary.

The paper studies the different forgetting patterns for the two popular post-training methods SFT and RL. The paper first establishes that SFT causes more severe forgetting than RL through extensive experiments ranging from instruction following, knowledge, and reasoning tasks. We further show that these forgetting patterns can be understood through the lens of forward/reverse KL that correspond to SFT and RL, and identify through ablation that **the use of on-policy data is the main reason for the lesser forgetting**. Based on this insight, we show that "approximately on-policy" methods such as iterative SFT can be a middle ground that has both the forgetting mitigation benefit and reduced cost.

All the reviewers recognize the soundness of paper, concluding that the experiments and ablations are comprehensive, rigorous, and the results are convincing. The reviewers raised several questions, which we responded in details in the rebuttal. Unfortunately, we have not received any follow-up before the system was frozen. We provide a brief summary below:

- **Reviewer toGA** raised a question regarding the practicality of Iterative-SFT. We clarified in detail that the method was based on the same assumptions as all the other baselines so the practicality is not a concern.

- **Reviewer rYjr** raised a question about our contribution compared with Tajwar et al., 2024. We clarified that Tajwar et al., 2024 does not tackle the issue of forgetting _at all_ and is orthogonal to our investigation. We also show that our investigations provide **actionable insights** (e.g, using approximately on-policy data) for practitioners to balance between forgetting mitigation and cost.

- **Reviewer uGfs** asked for evaluations beyond the settings in our paper. We provide extra experiments on chat eval showing a consistent trend with our reported results in the paper, validating the generality of our finding.


We have also answered other questions and concerns in details as well as updating the paper. Please refer to the individual response for more details. We once again thank all reviewers and chairs for their careful considerations.

---

### Meta-Review · Area_Chair_RSye · 2026-01-02

**Summary:**

This paper studies catastrophic forgetting in large language models during post-training and provides an extensive empirical comparison between supervised fine-tuning (SFT) and reinforcement learning (RL). The authors show that RL consistently forgets less than SFT across multiple model families (Llama, Qwen) and tasks, and attribute this behavior primarily to RL’s use of on-policy data. Through a simplified mixture-of-Gaussians analysis, the paper further connects forgetting behavior to the mode-seeking versus mode-covering properties of reverse and forward KL. As a practical implication, the authors propose using approximately on-policy data (e.g., Iterative-SFT) as a middle ground to mitigate forgetting with lower cost than full RL.

Reviewers broadly agreed that the paper is technically sound, clearly written, and empirically thorough, and appreciated the systematic experimental design and careful ablation of confounding factors. However, several reviewers raised concerns that **the conceptual contribution is incremental relative to existing literature**, and that the paper primarily consolidates and empirically validates insights that are already known or widely assumed in the RL and alignment communities. Despite a detailed and thoughtful rebuttal, these concerns remain only partially addressed, leading to a decision to reject.

**Reviewer Concerns:**

Concerns that have been addressed satisfactorily:
- In response to concerns about experimental rigor and coverage raised by Reviewers toGA and uGfs: the authors provided extensive evaluations across multiple model families, tasks (instruction following, knowledge, reasoning, safety, and conversational benchmarks), and metrics, demonstrating that the observed forgetting patterns are consistent and robust.
- In response to concerns about the practicality and assumptions of Iterative-SFT raised by Reviewer toGA: the authors clarified that Iterative-SFT relies on the same reward-model or verification assumptions as standard SFT and RL, and emphasized that Iterative-SFT is used primarily as an analytical tool rather than a proposed off-the-shelf replacement.
- In response to concerns about evaluation metrics and forms of forgetting raised by Reviewer uGfs: the authors justified their use of accuracy-based forgetting metrics by situating them within established continual learning practice, and added supplementary analyses on safety benchmarks and distributional drift.
- In response to concerns about applicability beyond reasoning and knowledge tasks raised by Reviewer uGfs: the authors added experiments on conversational benchmarks (AlpacaEval), showing consistent trends.

Concerns that have not been addressed satisfactorily:
- In response to concerns about novelty relative to prior work raised by Reviewer rYjr: while the authors clarified distinctions from closely related work (e.g., Tajwar et al., 2024), the central insight—that on-policy, mode-seeking updates mitigate forgetting—largely aligns with existing understanding in the RL and alignment literature, and the paper does not introduce fundamentally new theoretical frameworks or algorithms beyond this consolidation.
-  In response to concerns about the incremental nature of the contribution raised by Reviewer rYjr: the mixture-of-Gaussians analysis provides intuition but remains a stylized model that does not substantially advance theoretical understanding of forgetting in high-dimensional LLMs.
- In response to concerns about the broader impact on method design raised by Reviewer rYjr: given that on-policy methods (e.g., PPO, GRPO, DPO, rejection sampling) are already widely adopted in practice, the paper’s conclusions offer limited new guidance for future algorithm development beyond reinforcing existing best practices.
- In response to concerns about scalability and generality raised by Reviewer uGfs: experiments are limited to models up to 8B parameters, and it remains unclear whether the same conclusions hold for significantly larger models that are most relevant in current practice.

**Reviewer Scores:**

- Reviewer toGA: Marginally positive (6);
- Reviewer uGfs: Marginally positive (6);
- Reviewer rYjr: Marginally negative (4); raised concerns about incremental novelty and limited conceptual advancement.

Overall, reviewer opinions were mixed, with **persistent concerns about novelty and impact preventing consensus toward acceptance**.

---

### Decision · Program_Chairs · 2026-01-26

Reject